# Curcuminoid Chalcones: Synthesis, Stability, and New Neuroprotective and Sonosensitising Activities

**DOI:** 10.3390/ph16091331

**Published:** 2023-09-21

**Authors:** Dorota Olender, Małgorzata Józkowiak, Hanna Piotrowska-Kempisty, Katarzyna Sowa-Kasprzak, Lucjusz Zaprutko, Izabela Muszalska-Kolos, Ewa Baranowska-Wójcik, Dominik Szwajgier

**Affiliations:** 1Department of Organic Chemistry, Pharmaceutical Faculty, Poznan University of Medical Sciences, Grunwaldzka 6, 60-780 Poznań, Poland; kasik@ump.edu.pl (K.S.-K.); zaprutko@ump.edu.pl (L.Z.); 2Department of Toxicology, Pharmaceutical Faculty, Poznan University of Medical Sciences, Dojazd 30, 60-631 Poznań, Poland; malgorzata.jozkowiak@gmail.com (M.J.); hpiotrow@ump.edu.pl (H.P.-K.); 3Department of Pharmaceutical Chemistry, Pharmaceutical Faculty, Poznan University of Medical Sciences, Grunwaldzka 6, 60-780 Poznań, Poland; imuszals@ump.edu.pl; 4Department of Biotechnology, Microbiology and Human Nutrition, University of Life Sciences in Lublin, Skromna 8, 20-704 Lublin, Poland; ewa.baranowska@up.lublin.pl (E.B.-W.); dominik.szwajgier@up.lublin.pl (D.S.)

**Keywords:** curcuminoid chalcones, hybrids, NSAIDs, Claisen–Schmidt reaction, Steglich esterification reaction, sonodynamic therapy, biological activity, stress test

## Abstract

The primary purpose of this work was to design and obtain a series of curcuminoid chalcone–NSAID hybrid derivatives. The ester-type hybrid compounds with ibuprofen (**i**), ketoprofen (**ii**), and naproxen (**iii**) were obtained in two ways, using the Claisen–Schmidt reaction and the Steglich esterification reaction. The designed molecules were successfully synthesised, and FT-IR, MS, and NMR spectroscopy confirmed their structures. Moreover, the cytotoxic effect of the sonodynamic therapy and the anti-inflammatory, antioxidant, and anticholinergic properties of some curcuminoid chalcones and curcuminoid chalcones hybrids were evaluated. The curcuminoid chalcone derivatives showed promising neuroprotective activity as sonosensitisers for sonodynamic therapy in the studied cell lines. Additionally, the stability of the ester-type hybrid compounds with promising activity was determined. The RP-HPLC method was used to observe the degradation of the tested compounds. Studies have shown that structural isomers of ester-type hybrid compounds (**3ai**, **3bi**) are characterised by a similar susceptibility to degradation factors, i.e., they are extremely unstable in alkaline environments, very unstable in acidic environments, unstable in neutral environments, practically stable in oxidising environments, and photolabile in solutions and in the solid phase. These compounds maintain adequate stability in environment at pH 1.2 and 6.8, which may make them good candidates for developing formulations for oral administration.

## 1. Introduction

Designing and discovering new biologically active compounds and striving to find more selective synthesis strategies are among the critical tasks of medicinal chemistry. Molecular hybridisation is an effective and beneficial approach to developing new, multifunctional compounds specifically obtained with selected, active subunits by appropriate chemical reactions [1]. The combination of two or more biologically active fragments into one molecule very often has a beneficial effect on the pharmacokinetic parameters of the compound, including improving the solubility and oral bioavailability, enabling it to act in the right target place and at the same time, which has a positive impact on the therapeutic prognosis. These chalcones are recognised as a privileged scaffold for incorporating different molecules or pharmacophores with various activities [2]. Due to their diverse pharmacological activity, chalcones and their derivatives are important in medicinal sciences [3,4,5,6]. The pharmacological action of chalcones depends on the presence of the functional groups, halogens, and aromatic or heteroaromatic rings, as well as their placement in the chalcone structure. For example, 2′-hydroxychalcone inhibited nuclear factor-ĸB activation and blocked TNF-α- and LPS-induced adhesion of neutrophils to human umbilical vein endothelial cells [7]. 2′,5′-Dihydroxy-4-chloro-dihydrochalcone was also shown to inhibit LPS-stimulated nitric oxide synthesis and cyclooxygenase-2 activity [7]. 3,4,2′,4′-Tetrahydroxychalcone exhibited a potent inhibitory effect on H_2_O_2_-induced hemolysis due to an antioxidant effect [7]. Compounds in which electron-releasing groups such as methoxy and hydroxy are present exhibit greater antibacterial activity than other groups. Compounds with pharmacophores such as chlorine, dichlorine, bromine, and fluorine substitutions display increased antifungal activity [8]. According to many studies and biological tests, compounds based on chalcone scaffolds are promising agents in the treatment of different diseases, and some of them, such as metochalcone (choleretic drug), hesperidin methyl chalcone (vascular protective), and sofalcone (antiulcer and mucoprotective), have been approved for clinical applications. Due to their structure, with the possibility of modifications and pharmacodynamic properties, these compounds are also promising in the drug discovery of central nervous system diseases, including Alzheimer’s disease (AD) [9].

An important factor influencing the biological activity of chalcones is the double bond of the α,β-unsaturated carbonyl, as well as the planar structure geometry [10,11]. The possibilities of modifying the chalcone structure make it possible to obtain potential compounds with anticancer properties. The hybridisation of chalcones with anticancer pharmacophores is favourable for their activity. For example, introducing an azole into a molecule of these compounds significantly increases their biological properties, and it is a promising method for developing novel anticancer agents [12,13]. The anticancer properties of chalcones are mainly influenced by the presence of substituents on the chalcone molecule’s two aryl rings and the structural modifications of the aryl rings. Significantly, a fluorine atom and hydroxy and methoxy groups in the specific positions on the phenyl ring contribute to the anticancer activity of synthetic chalcones [14].

Nowadays, in the case of cancer diseases, there are different treatment strategies regarding the use of CAR T cells, cancer vaccines [15], molecular agents enhancing the effect of radiotherapy, and the application of high- and low-intensity focus ultrasound [16,17].

Sonodynamic therapy (SDT) represents a new approach that seems to have become a promising treatment, especially in less accessible places, providing the possibility of noninvasively eradicating solid tumours site-directedly. This method employs compounds that become cytotoxic after exposure to low-intensity ultrasound [18]. Essentially, both aspects, i.e., sensitisation and ultrasound exposure, are harmless, and cytotoxic events occur when combined [19]. The literature data [20,21] support that SDT could significantly benefit treating a wide range of cancers. In addition to demonstrating efficacy in a wide range of cell lines, in vitro studies have also shown that SDT can induce apoptosis or programmed cell death. However, the extent to which this occurs depends on the sonosensitiser used, the ultrasound’s characteristics, and the target’s nature. The mechanisms by which ultrasound enhances the action of chemotherapeutic drugs are varied. It is generally recognised that sonodynamic therapy refers more specifically to sensitiser-dependent sonochemical or sonophotochemical events in an acoustic field that lead to cytotoxicity. There is a clear link between ultrasound exposure, the presence of the sonosensitiser, and the generation of ROS. There is little doubt that ROS are involved in SDT-based mechanisms. The uncertainty relates to the method of their generation. It is also clear from the literature that the nature of ROS generated from ultrasound exposure varies with the sensitiser’s character [19].

Many sonosensitisers used in sonodynamic therapy were initially used as photosensitisers, the most common of which are the porphyrins, acridine orange, methylene blue, fluoroquinolone antibiotics, and natural products such as curcumin [20,21,22,23]. In particular, phthalocyanines derived from chalcones have many applications, including photodynamic therapy [24,25] (Figure 1).

Sonosensitisers are an indispensable component of SDT. Ideally, sonosensitisers should be highly sonosensitive, nontoxic in the absence of ultrasound, have specific accumulation at the tumour site, and have the ability to be excreted from the body within a short period [26]. The sound sensitisers are divided into four categories: porphyrin compound, xanthone compound, antitumour drug, nonsteroid anti-inflammatory drug, and other sound sensitisers [27]. The results of many experiments show that a significant sonodynamically induced antitumour effect was demonstrated with pheophorbide a [28], photofrin II [29], 5-fluorouracil [2], functionalised carbon nanotubes [30], or xanthone compounds [27]. The studies show that 5-ALA mediated SDT plays a role in the apoptosis of cells [31]. Moreover, other works suggest that the combinational use of sonodynamically active several nonsteroidal anti-inflammatory drugs [20,32] and ultrasound has significant anticancer effects [26]. Our recent study showed that a curcumin-triterpene type hybrid is an effective sonosensitiser for sonodynamic therapy in oral squamous cell carcinoma [33].

In light of the present knowledge of the meaning of the imbalance between prooxidant and antioxidant homeostasis and the role of free radicals in the pathogenesis of a majority of multifactorial diseases such as atherosclerosis, hypertension, Alzheimer’s and Parkinson’s diseases, cancer, and inflammatory conditions, this study evaluated the cytotoxic effect of sonodynamic therapy and the anti-inflammatory, antioxidant, and anticholinergic properties of some curcuminoid chalcones and curcuminoid chalcone hybrids.

## 2. Results and Discussion

### 2.1. Chemical Parts

#### Synthesis of Curcuminoid Chalcone–NSAID Hybrid Derivatives

From the literature [32,34], it is known that nonsteroidal anti-inflammatory drugs such as tenoxicam and piroxicam, which have a conjugated double bond in the molecule, increase the antitumour effects of ultrasound by increasing the production of singlet oxygen and other active oxygen species. These findings indicated that the enhanced antitumour effect is attributed to the sonodynamic effect of the ultrasound in combination with the NSAIDs. Furthermore, our recent study showed that a curcumin–triterpene-type hybrid is an effective sonosensitiser for sonodynamic therapy in oral squamous cell carcinoma [33]. According to these results, in this study, we decided to conduct structural modifications by functionalising curcuminoid chalcones directly to obtain ester-type hybrid derivatives of curcuminoid chalcones.

The primary structural transformation planned to afford curcuminoid chalcone hybrid derivatives was the reaction of the hydroxyl group in the chalcone skeleton with the selected free carboxylic moiety of anti-inflammatory agents. The synthesis of ester-type hybrid derivatives obtained in this way was carried out in two ways. First, as substrates, curcuminoid chalcones were used. These compounds were synthesised by the Claisen–Schmidt reaction according to the procedure previously reported in the literature [35]. The starting reagents in the synthesis of curcuminoid chalcones (**3a**–**b**) were acetophenone and two aromatic aldehydes, i.e., 4-hydroxy-3-methoxybenzaldehyde (vanillin) and 3-hydroxy-4-methoxybenzaldehyde (isovanillin). The reactions were carried out in an ethanolic solution of aromatic aldehyde and acetophenone, to which sodium hydroxide (NaOH) aqueous solution (50%) was added at room temperature. Compounds **3a** and compound **3b** were subjected to functionalisation via transformations of the phenolic groups of selected curcuminoid chalcones towards the corresponding derivatives, namely esters. To obtain ester-type products of curcuminoid chalcones, the appropriate curcuminoid chalcone was reacted with a selected drug from the group of nonsteroidal anti-inflammatory drugs. Among them, ibuprofen (**i**), ketoprofen (**ii**), and naproxen (**iii**) were applied as nonsteroidal anti-inflammatory particles (Figure 2).

Second, curcuminoid chalcone–NSAID hybrid derivatives (**3ai**–**3biii**) were obtained in the reactions of appropriate ester-type connections of aromatic aldehyde (**2ai**–**2biii**) with acetophenone via the Claisen–Schmidt reaction. The reactions were carried out in an ethanolic solution of vanillin (**2a**) or isovanillin (**2b**) and acetophenone, to which sodium hydroxide (NaOH) aqueous solution (50%) was added at room temperature. The ester-type connection of aromatic aldehydes (**2ai**–**2biii**) was obtained due to the condensation of compound **2a** or compound **2b** with equimolar quantities of NSAIDs.

Two chemical directions of synthesis of curcuminoid chalcone–NSAID hybrid derivatives are presented in Figure 1.

Both aromatic aldehydes with a hydroxy group (**2a** and **2b**) and curcuminoid chalcones (**3a** and **3b**) were reacted with appropriate NSAID compounds (**i**–**iii**) in the presence of dicyclohexylcarbodiimide (DCC) and 4-(dimethylamino)pyridine (DMAP) reagents. DCC is a carboxylic-acid-activating and -coupling agent. At the same time, the strongly nucleophilic DMAP acts as an acyl transfer reagent in the Steglich esterification reaction [36,37], which is a beneficial process in the conversion. DCC reacts immediately with the water resulting from the reaction and forms dicyclohexylurea (DHU), which is insoluble in the reaction medium and precipitates as a white solid. Therefore, it is essential to carry out the reaction in anhydrous conditions. A high DMAP addition rate is also crucial for the efficient formation of esters in cases where the catalytic amount does not yield the product expected. Both in the esterification reaction of hydroxy aromatic aldehydes or curcuminoid chalcones, the molar ratio of aromatic aldehydes or curcuminoid chalcones, NSAIDs, DCC, and DMAP reagents used in the reaction process was 1:1.2:1.6:0.8. All reactions were carried out in an anhydrous dichloromethane solution. The reagents were mixed and then stirred at room temperature for up to four hours to achieve complete or almost complete conversion of the appropriate substrate, i.e., the vanillin and isovanillin or curcuminoid chalcones. In the above two types of reactions, from different substrates, final hybrid connections of curcuminoid chalcone–NSAID (**3ai**–**3biii**) as ester-type derivatives were obtained (Figure 3).

Further isolation and purification by flash column chromatography resulted in the desired pure esters. FT-IR, MS, ^1^H, and ^13^C NMR data confirmed the structures of the above hybrid compounds (**3ai**–**3biii**) (Figure 3). While controlling the runs of the two types of reactions using the TLC method, the formation of a by-product was noted. The spectral analysis of the isolated by-product showed this product’s appearance by competing a DCC reaction with the appropriate NSAID. This fact, unfortunately, reduced the yield of ester derivatives from the obtained curcuminoid chalcones.

The FT-IR spectra of the prepared curcuminoid chalcone–NSAID hybrids (**3ai**–**3biii**) clearly show the existence of the characteristic stretching vibration bands. In the spectrum of compound **3ai**, the stretching vibration band of the ketone moiety disappears at 1661 cm^−1^ in propenone and 1749 cm^−1^ in the ester moiety. The presence of molecular peaks was found in the EI-MS mass spectra of the obtained compounds, consistent with the calculated molar mass values for these compounds. These compounds fragment in a specific way. The basic fragmentation pathways of curcuminoid chalcone–NSAID hybrids are obtained via ester bond breakage. Additionally, there is a loss of the phenyl cation (C_6_H_5_^+^) and a loss of C≡O^+^. In the exemplary case of compound **3biii**, the peaks at m/z = 213 and at m/z = 253 are due to the formation of ions corresponding to the naproxen and chalcone moieties as a result of the ester bond breaking. Next, fragmentation is associated with the loss of C≡O^+^ from the naproxen ion and the formation of an ion at m/z = 185. The subsequent fragmentation is a consequence of the loss of the methoxy group (Figure 2).

The signals of protons and carbons of curcuminoid chalcone–NSAID hybrids were assigned based on the results of the ^1^H and ^13^C NMR spectra. Generally, the ^1^H NMR spectra of the obtained compounds show the trans configuration of vinyl protons in the propenone, which links the aromatic rings. For example, two doublets are visible in the spectra of compound **3bi** at 7.73 ppm and 7.36 ppm, with a coupling constant of 15.50 Hz. Protons derived from the methoxy group in singlets occur at 3.95 ppm. Finally, the signals between 8.09 and 7.80 ppm are attributed to aromatic protons of the acetophenone ring. In the ^13^C NMR spectra, characteristic signals of the products were obtained, including those originating from the carbonyl group in the aliphatic chain and in the ester bond. In the spectrum of compound **3bi**, these signals occurred at 186.30 and 173.53 ppm, respectively. Signals from the carbons in the methoxy group were present at 55.89 ppm.

### 2.2. Biological Activity

Various biological activities of chalcone derivatives, such as anticancer, anti-inflammatory, antituberculosis, antimalarial, antioxidant, antimicrobial, antidiabetic, and antiviral, and cancer prevention activities, have been reported [4,8]. This means that there are a wide range of chalcones with a wide number of biological activities. Our study has reported that some curcuminoid chalcones showed promising anticancer potential in the human colon carcinoma (Caco-2) cell line. The obtained results suggest that one compound tested increases the metabolism or proliferation of the Caco-2 cell line, while other compounds showed significant toxicity in the Caco-2 cell line. The biological study confirmed that anticancer activity is an important factor of hydroxy and methoxy groups in at least one aromatic ring. Moreover, from the literature, it is known [32] that nonsteroidal anti-inflammatory drugs such as tenoxicam and piroxicam, which have a conjugated double bond in the molecule, increase the antitumour effects of ultrasound by increasing the production of singlet oxygen and other active oxygen species. These findings indicated that the enhanced antitumour effect is attributed to the sonodynamic effect of the ultrasound in combination with the NSAIDs. Chalcone is also one of the privileged scaffolds with diverse biological functions that are being extensively studied for AD.

Taking the effect of the structure on the biological activity of derivatives based on the chalcone skeleton in this study, the cytotoxic effect of sonodynamic therapy; the anti-inflammatory, antioxidant, and anticholinergic properties of some curcuminoid chalcones and curcuminoid chalcone hybrids with ibuprofen; and also ibuprofen as their substrate were evaluated (Figure 4). The compounds tested were numbered, as shown in Figure 4. Moreover, the stability of new hybrid ester-type compounds was evaluated.

#### 2.2.1. Sonodynamic Studies

According to the literature data [32] and biological results obtained in our study [33,35], we decided to select two curcuminoid chalcones also synthesised in previously reported work [35], a hybrid connection of curcuminoid chalcone and ibuprofen (Figure 4). The compounds used for the sonodynamic study to evaluate anticancer activity were marked **1**, **2**, **3**, and **4**. In earlier experiments, 3-(4-hydroxy-3-methoxyphenyl)-1-phenylprop-2-en-1-one (**1**) and 1,3-bis(4-hydroxy-3-methoxyphenyl)-1-phenylprop-2-en-1-one (**3**) increased ROS production in the Caco-2 cell line in both studied time intervals (24 and 48 h) [35]. As mentioned in the introduction, ROS are involved in SDT-based mechanisms, and the only degree of uncertainty relates to how those are generated. The increase in ROS levels can initiate cell death via caspase-dependent and caspase-independent apoptosis, as well as necrosis [38]; compound **1** strongly induced ROS expression and toxicity, as measured by resazurin and LDH. In addition, this compound reduced the expression of antioxidant enzymes that could break down ROS, which is good information in the case of cancer [35]. In the sonodynamic method, the synergism of the actions of chemical compounds and ultrasounds is used as a new approach in cancer therapy.

The cytotoxic effect and sonosensitising properties of selected synthesised chalcone derivatives were evaluated on human SCC-25 tongue and FaDu throat cancer cell lines using the MTT test (Figure 5 and Figure 6).

The viability rates of SCC-25 and FaDu cells after US exposure were inhibited by ~13% and ~32%, respectively. We observed significantly reduced viability of the FaDu cell line treated with compound **1** combined with US compared to treatment with compound **1** alone. The cytotoxic activity of compound **2** was similar to that of compound **1**. The viability rates of the FaDu cell line treated with compound **3** alone and the combination of compound **3** and US were profoundly inhibited (~65% and ~80%, respectively) compared to untreated cells (Figure 5).

There was no significant difference in SCC-25 cells treated with **1** alone compared to the control (Figure 6).

Conversely, the treatment with compound **1** and the US reduced the viability of SCC-25 cells by ~40% compared to untreated ones. Similarly, the cytotoxicity of compound **2** and compound **3** was lower than that of compound **2** and compound **3** combined with US; however, only by ~10%. The cytotoxic activity of compound **4** was similar to that of compound **1** and compound **2**. The cytotoxic activity and viability of SCC-25 and FaDu cells treated with compound **4** were similar to the control. There was no significant difference in SCC-25 and FaDu cells treated with the combination of compound **4** and US. The percent of the viability of these cells was significant.

The results of sonodynamic tests show that the ester-type product of curcuminoid chalcone (**2**) in combination with US shows a much stronger effect than the compounds from which it was formed, i.e., compounds **1** and **4**. This is particularly visible in the case of the SCC-25 cell line; we observed significantly reduced viability of the SCC-25 cell line treated with compound **2** combined with US.

However, in the case of derivatives of curcuminoid chalcones, the activity of compound **3** is definitely more significant than that of derivative **1**, which can be explained by the presence of additional substituents in the A ring. The conducted studies allowed for the initial estimation of the cytotoxic effects of synthetic chalcones and ultrasounds in head and neck cancer cell lines. Compounds **2** and **3** showed promising anticancer potential in the studied cell line, and these compounds could be effective sonosensitisers for sonodynamic therapy.

#### 2.2.2. Anti-Inflammatory Activity

##### Inhibition of Cyclooxygenase-2 (COX-2)

The presence of COX-2 in cells is associated with the inflammatory process and degenerative or neoplastic changes. COX-2 inhibitors directly target cyclooxygenase-2 [39].

There is considerable interest in looking for compounds that inhibit COX-2 due to the possibility to retard or prevent the development of colon cancer, which is often characterised by COX-2 expression. Moreover, another potential target for COX-2 inhibitors is AD, which is characterised by inflammatory processes in the brain that are associated with increased COX-2 expression. Epidemiological surveys (e.g., gathered in excellent reviews by Gasparini et al. [40] and Maccioni et al. [41]) point out delayed expression or slow progression of the disease caused by the use of NSAIDs, especially at early stages of amyloid plaque formation. Despite the understanding of multiple molecular pathways in the pathophysiology of AD, novel treatment agents with a possible role in modifying the disease activity still need to be discovered.

The inhibition of the COX-2 enzyme by curcuminoid chalcones and ibuprofen as a reference substance is shown in Table 1.

The results of our study indicate that curcuminoid chalcones **1, 3,** and **7** showed the ability to inhibit COX-2. Among them, the strongest inhibition of COX-2 was exerted by **7**. Interestingly, the curcuminoid chalcone–NSAID hybrids were not able to inhibit COX-2.

The obtained data indicate that the substances tested can attenuate the inflammatory response induced by COX-2, which can be beneficial in the context of neurodegenerative diseases.

##### Effect on Antioxidant Enzyme Activity

Reactive oxygen species (ROS) are harmful to our health, and superoxide dismutase (SOD), catalase (CAT), and glutathione peroxidase (GPx) are the major “antioxidant” enzymes that are important factors during the production and penetration of free radicals and their penetration into cells and tissues. Enzymatic antioxidant mechanisms based on SOD2 were investigated in our recent work on selected curcuminoid chalcones using the qRT-PCR method [35]. Among the tested compounds, compound **6** significantly affected the expression of the SOD2 gene (a decrease of 65.20%) compared to the control [35].

We decided to extend the study by measuring the compounds’ ability to inhibit glutathione reductase (GR) and GPx activities, the human cells’ main endogenous enzymatic defence systems. Therefore, we checked the influence of some curcuminoid chalcones on GR and GPx using in vitro spectroscopic methods (Table 2). Moreover, the effects of curcuminoid chalcones on CAT activity were estimated (Table 2).

All antioxidant enzymes were inhibited by the curcuminoid chalcones **1** and **7.** Both enzymes (GP and GPx) were inhibited by the curcuminoid chalcones **1** and **7**, and additionally the curcuminoid chalcone–NSAID hybrid **2**. The results show that the most potent inhibitory effect was demonstrated by compound **7**, which inhibited both GR, GPx, and CAT (41.8 ± 3.1%, 32.4 ± 2.8%, and 88.1 ± 4.0%, respectively). Quite potent inhibition against GR was presented by nonsubstituted chalcone **5**, which did not inhibit the GPx at all. Interestingly, the curcuminoid chalcone–NSAID hybrid **6** did not inhibit the GR, while its isomer **2** presented quite potent inhibition.

##### Hydroxyl Radical Antioxidant Capacity (HORAC)

A hydroxyl radical antioxidant capacity (HORAC) assay is a classic tool for measuring the antioxidant capacity levels of different substances.

In our work, we estimated the curcuminoid chalcones’ capacity to block the fluorescent probe’s radical hydroxyl oxidation until the antioxidant activity in the sample was depleted. The sample’s antioxidant capacity correlated to the fluorescence decay curve and was compared to a gallic acid antioxidant calibration (Table 3).

The curcuminoid Chalcone–NSAID hybrids (**2**) and (**6**) showed antioxidant activity in the HORAC test.

#### 2.2.3. Anticholinergic Activity

Neurodegenerative diseases are associated with the progressive damage of neurons. During the development of these diseases, there is a loss of nerve cells and a deficiency of acetylcholinesterase (AChE), serotonin, and norepinephrine levels. The ability to inhibit acetyl- (AChE) and butyrylcholinesterase (BChE) activities is seen, among others, in organophosphorus compounds, carbamate insecticides, phenoxyacetic pesticides, and hydrogen peroxide, as well as aliphatic ketones with the carbons C4–C9, e.g., 2,5-hexandione. AChE and BChE show differences in their susceptibility to inhibitors, behaviour in situations involving excess substrate, or substrate binding specificity. Unlike, other esterases, both enzymes hydrolyse choline esters at very high speeds [42]. Thus, in our work, the modified Elman’s method was used to investigate the anti-AChE and anti-BChE effects of curcuminoid chalcone.

Our results indicated the excellent activity of the tested compounds (Table 4).

The enzymes were inhibited by all compounds tested. Our results indicated the very potent activity of the compounds tested. The curcuminoid chalcone–NSAID hybrids **2** and **6** inhibited AChE and BChE more than their substrates **1** and **7**. The greatest inhibition of AChE and BChE was shown by compound **7**.

#### 2.2.4. Estimation of Stability of Ester-Type Hybrid Compounds

##### Stress Test

Considering the results of the sonodynamic tests, we decided to determine the stability of two structural isomers (marked as **2** and **6**) to know the relationship between the structure and the stability of the hybrid compounds. This study allowed us to determine the susceptibility of hybrid compounds to chemical degradation and sensitivity to external factors (pH, oxidising agents, and light), which is especially necessary to predict its interaction in the pharmaceutical phase.

According to the ICH (International Conference on Harmonization of Technical Requirements for Registration of Pharmaceuticals for Human Use) guidelines, each chemical substance should be tested for its sensitivity to degradation factors, such as hydrolytic agents (acid, alkaline, and neutral environments), oxidising agents, and degradation under the influence of light. The analysis of the compound’s susceptibility to decomposition allows for the development of appropriate storage conditions, both in the solid phase and in solution, and the possible optimisation of the synthesis process by minimising these factors, affecting the efficiency (yield) of the synthesis.

Several tests were conducted using the ICH guidelines to assess the compound’s stability. In the stress test, two structural isomers of ester-type hybrid compounds were chosen to estimate the structure’s relationship to the compound’s stability. The tested substances were subjected to hydrolysis–solvolysis in neutral, acidic, and alkaline media. As solvents, water, hydrochloric acid, and sodium hydroxide were used in appropriate concentrations.

Compound **2** and its isomer **6** were analysed using a stress test, considering the effects of NaOH, HCl, water, H_2_O_2_, and light on their decomposition. Reversed-phase HPLC was used to analyse the degree of degradation. Chromatograms were developed with DAD detection at 254 nm, 308 nm (λ_max_ for **2**), and 340 nm (λ_max_ for **6**). The main peaks of the tested compounds were observed on the chromatograms at 12.6 min (**2**) and 11.5 min (**6**). Under the same conditions, the ibuprofen retention time (Rt) was 2.9 min. Both compounds were completely degraded in HCl 0.1 M in the acidic medium at 90 °C. At the same time, in milder conditions (0.01 M HCl, 40 °C, 8 h), their decomposition rates were at the level of 44% (**2**) and 18% (**6**) (Table 5). This may suggest the influence of isomerism on the stability of the structure of the analysed compound in an acidic condition.

However, according to the ICH guidelines, both compounds should be classified as very unstable. At the same time, the chromatograms recorded at 254 and 308 nm (Figure 7) showed peaks with Rt = 6.3 (the impurity observed in the t_0_ chromatograms) and 11.2 min (a potential decomposition product) for **2** solutions. On the other hand, no additional peak was observed in the chromatograms for the compound **6** solutions, which may have been due to insufficient degradation of the compound (18%).

Both compounds were completely degraded in NaOH (0.01 mol/dm^3^) at room temperature after only 2 h of exposure (Table 5). According to the ICH guidelines, they should be classified as extremely unstable compounds. In addition, during their decomposition at 90 °C, a temporary appearance of yellow to orange colour was observed, which disappeared after 24 h. The chromatograms recorded at 254 nm (Figure 8) showed peaks at Rt = 2.2 (main peak) and 5.4 min for the **2** solutions and 2.2 (major peak), 2.6, 3.1, and 3.2 min (one-third of the main peak) for the **6** solutions.

In a neutral environment, both compounds, based on the observed 24% of their degradation at 90 °C (Table 1), were classified as unstable compounds. At the same time, on the chromatograms recorded at 254 and 308 nm (Figure 9), peaks at Rt = 6.3 (contamination observed in the t_0_ chromatograms) and 11.2 min (potential decomposition product) for the **2** solutions and 10.7 min (potential decomposition product) for the **6** solutions were observed.

The peaks with Rt values of 6.3 (Figure 7a,c) and 4.4 min (Figure 7b,d) observed in the chromatograms are impurities of compounds **2** and **6**, respectively. None of the chromatograms showed a peak characteristic of ibuprofen—a potential product of their hydrolysis. However, this fact should be explained by its high sensitivity to decomposition under the influence of temperature and in the presence of water. Water solutions of ibuprofen show the greatest stability at pH 5–7 and at temperatures below 30 °C [43].

Therefore, it should be considered that under the tested stress conditions, observing the ibuprofen peak on the chromatograms of the tested compounds is impossible. On the other hand, the small difference between the Rt values of the observed products, 11.2 min and 10.7 min, and their similarity in the UV spectra (Figure 10) in the chromatograms of the **2** and **6** solutions, respectively, suggest that the observed peaks are of positional isomers, i.e., 3-(4-hydroxy-3-methoxyphenyl)-1-phenylprop-2-en-1-one and 3-(3-hydroxy-4-methoxy-phenyl)-1-phenylprop-2-en-1-one.

This may prove that in neutral and acidic conditions, we are dealing with the hydrolysis of the ester group of the compounds tested. On the other hand, in an alkaline medium, further degradation of the resulting products occurs.

The tested compounds were insensitive to oxidising agents (Table 5). In solutions of hydrogen peroxide (1–10%), no signs of degradation were observed after 24 h (Figure 11). Therefore, they should be classified as a group of compounds practically stable in an oxidising environment.

However, as a result of the action of light, both compounds degraded completely (Table 6, Figure 12). The study was conducted in methanol solutions, irradiated in quartz cuvettes, and in solutions protected from light (aluminium foil). After exposure to a dose of 1.2 × 10^6^ lux·h, significant decomposition of compound **2** was observed in the sample protected from light (68%), which may indicate its sensitivity to temperature in this condition. During irradiation, the temperature in the Suntest chamber was 35 °C. It was, therefore, concluded that both compounds are photolabile in solutions. Only in the chromatogram of the degraded compound **6** recorded at 254 nm was a product with an Rt of 5.8 min observed (Figure 8b). The visible peaks with Rt values of 6.3 (Figure 12a,c) and 4.4 min (Figure 12b,d) were impurities of compounds **2** and **6**, respectively.

The study of the effect of light on the decomposition of the compounds tested in the solid phase consisted of the analysis of changes in the IR spectrum before and after the irradiation of tablets (with KBr) with doses of 1.2 × 10^6^ and 6.0 × 10^6^ lux·h. The spectra obtained were compared for consistency with that of the tablet before irradiation (Table 2). In the case of compound **2**, the IR spectrum (Figure 13) shows clear changes in the range of 1800–1600 cm^−1^. Bands 1662.7 cm^−1^ and 1700 cm^−1^ disappear and band 1757.2 cm^−1^ appears. In the case of compound **6**, these changes are not so visible (Figure 14), which may indicate its greater resistance to light. Nevertheless, in this case, changes were observed within the 1658.8 cm^−1^ band. The observed changes may qualify both compounds as photolabile in the solid phase.

##### Stability Test in Dissolution Solutions

Considering the possibility of developing an oral formulation of a pharmacologically active compound, it was decided to analyse the stability of the tested compounds in standard solutions used to study the release of the active substance from solid drug forms. For this purpose, 0.1 M of hydrochloric acid (pH 1.2) and phosphate buffer at pH 6.8 was selected. Due to the effect of precipitation of the tested substances observed in aqueous solutions, a solubiliser (1% Tween 80) was added. The test was carried out at 37 °C. Both compounds were so stable under these conditions that it was impossible to interpret their kinetic degradation mechanism or establish an appropriate equation. Within five days of incubation, compound **2** degraded only 15–19% and **6** 11–17% (Table 7). Such slight losses do not allow for a comparative analysis of the stability of both compounds. After this time, the precipitation process was observed. It can, therefore, be assumed that both compounds will maintain adequate stability in the gastrointestinal tract under the conditions of oral administration.

However, after five days of incubation, the chromatograms recorded at 254 nm showed their decomposition products with Rt values of 11.2 min and 13.9 min for the **2** solutions and 10.8 and 13.9 min for the **6** solutions, both at pH 1.2 and 6.8 (Figure 15).

Products with Rt values of 11.2 and 10.8 min were also observed in the stress test in the acidic and neutral conditions (Figure 7 and Figure 9), while the product with an Rt of 13.9 min was not observed in the chromatograms recorded at λ_max_ (i.e., 308 nm for **2** and 340 nm for **6**) (Figure 16).

The analysis of the spectra characterising the peaks of the compounds tested and the decomposition products (Figure 17) indicated the structural similarity of the products at Rt = 13.9 min. At the same time, the differences in the Rt values of products of 11.2 and 10.8 min may confirm that they are isomers of each other. The chromatograms did not observe the peak for ibuprofen (2.9 min). In contrast, peaks of 13.9 were observed only in solutions containing Tween 80, which may indicate a different degradation mechanism than aqueous solutions. The presence of Tween 80 could also have a stabilising effect on the solutions tested.

## 3. Materials and Methods

### 3.1. Solvents and Chemicals

An aqueous solution of NaOH (50%), acetophenone, 4-hydroxy-3-methoxy-benzaldehyde (vanillin), 3-hydroxy-4-methoxybenzaldehyde (isovanillin), DCC, DMAP, NSAIDs (ibuprofen, ketoprofen, naproxen), and solvents (ethanol, dichloromethane, ethyl acetate, and n-hexane) from Aldrich (St. Louis, MO, USA), Fluka (Buchs, Switzerland), Chempur (Piekary Śląskie, Poland), and POCh S.A. (Gliwice, Poland) were used. The acetonitrile for high-performance liquid chromatography (HPLC), release buffer solution, and phosphate buffer (pH 6.8) were from Chempur (Piekary Śląskie, Poland). The Tween 80 was from Sigma-Aldrich (Steinheim, France).

All other chemicals of the highest purity were commercially available, and demineralised water was used in the tests.

### 3.2. Instrumental Analysis

The melting points of all compounds used in this study were determined on a Boetius apparatus and were uncorrected. The IR spectra were recorded using a Nicolet iS50 FT-IR spectrometer (Thermo Scientific, Waltham, MA, USA). The ^1^H and ^13^C NMR spectra were recorded using an NMR Varian VNMR-S 400 MHz spectrometer at 400 and 100 MHz, respectively (Agilent Technologies, Santa Clara, CA, USA). The chemical shifts were expressed in parts per million (ppm) relative to tetramethylsilane (TMS) as an internal standard, using CDCl_3_ as the solvent. Coupling constants (*J*) are expressed in Hertz (Hz). Signals are labelled as follows: s, singlet; d, doublet; dd, double doublet; t, triplet; m, multiplet. The MS spectra were recorded on a Bruker 320MS/420GC spectrometer apparatus (Bruker Corporation, Billerica, MA, USA) using the electron impact technique (EI), operating at 75 eV. The progress of reactions and the purity of products were checked using the TLC method on silica gel plates (DC-Alufolien Kieselgel 60 F_254_ from Merck, Darmstadt, Germany). Hexane and ethyl acetate (2:1 and 9:2 *v*/*v*) were used as the eluents. The TLC spots on the plates were observed in UV light (λ = 254 nm). Silica gel 60 (63–200 μm particle size, Merck) was used for the column chromatography. The crude reaction products were purified by means of a crystallisation process or flash column chromatography using a mixture of hexane and ethyl acetate (2:1, *v*/*v*).

### 3.3. General Synthesis Procedure for Curcuminoid Chalcones ***3a*** and ***3b***

The aqueous solution of 50% NaOH (5 mL) was added dropwise into a solution of 3 mmol (0.36 g) of acetophenone in ethanol (20 mL). A total of 3 mmol of vanillin or isovanillin (0.46 g) in an ethanolic solution (5 mL) was introduced into the mixture. The mixture was stirred at room temperature for 48 h. The mixture was then poured into ice water and neutralised with 10% HCl to produce precipitates. The solid precipitates were filtered, washed with water, and crystallised with methanol to yield the final compounds. The aquatic solution was extracted with dichloromethane when the solid was not formed. The combined organic layer was washed with 10% HCl and then with water. The organic layer was dried over the anhydrous MgSO_4_ and the solvent was removed under reduced pressure. The crude solid was purified via column chromatography using hexane/ethyl acetate (2:1, *v*/*v*) as an eluent.

*E*-3-(4-hydroxy-3-methoxyphenyl)-1-phenylprop-2-en-1-one **(3a)**. Yield: Yellow crystals (78.74%), m.p.: 88–90 °C, R_f_ = 0.35 (hexane/ethyl acetate, 2:1, *v*/*v*). Spectral data in accordance with literature data [35].

*E*-3-(3-hydroxy-4-methoxyphenyl)-1-phenylprop-2-en-1-one **(3b).** Yield: Orange crystals (74.80%), m.p. 89–90 °C. R_f_ =0.25 (hexane/ethyl acetate, 2:1, *v*/*v*). Spectral data in accordance with literature data [35].

### 3.4. General Synthesis Procedure for Curcuminoid Chalcone–NSAID Hybrid Derivatives (***3ai****–**3biii***) in the Reaction of Curcuminoid Chalcones with NSAIDs

Curcuminoid chalcone **3a** or **3b** (0.5 mmol), DCC (1.03 g, 0.8 mmol), DMAP (5.0 mg, 3 mmol), and an anti-inflammatory drug (ibuprofen (**i**), naproxen (**ii**), or ketoprofen (**iii**) (0.6 mmol)) were mixed at 0 °C and then stirred at room temperature in dry CH_2_Cl_2_ (12 mL) for 4 h. The resulting precipitate of dicyclohexylurea was filtered. The organic phase was washed with 5% aqueous hydrochloric acid, 5% sodium bicarbonate, and then with water. The organic layer was dried over the anhydrous MgSO_4_ and the solvent was removed under reduced pressure. The crude new esters (**3ai**–**3biii**) obtained were crystallised or purified by column chromatography on silica gel using a mixture of hexane/ethyl acetate (2:1, *v*/*v*) as an eluent.

2-Methoxy-4-(3-oxo-3-phenylprop-1-en-1-yl)phenyl-2-(4-isobutylphenyl)-propionate **(3ai)**

Yield: Light yellow needles (45.24%), m.p.: 81–82 °C, R_f_ = 0.35 (CHCl_3_:methanol, 9:2, *v*/*v*).

FT-IR (υ cm^−1^): 2949 (C-H), 2915 (C-H), 2849 (C-H), 1749 (C=O), 1661 (C=O), 1607 (C=C), 1597 (C=C), 1509 (C=C), 955 (=C-H), 777 (C-H), 696 (C-H); EI-MS, m/z (%): 442 (52) M^+^, 412 (100), 374 (2), 253 (2); ^1^H NMR (400 MHz, CDCl_3_) (δ[ppm]): 8.02–7.98 (m, 2H, ArH), 7.73 (d, *J* = 15.70 Hz, 1H,=CH), 7.62–7.55 (m, 1H, ArH), 7.53–7.48 (m, 2H, ArH), 7.31 (m, 2H, ArH), 7.37 (d, *J* = 15.50 Hz, 1H, CH=CH), 7.22–7.19 (dd, *J* = 8.18 Hz, *J* = 1.94 Hz, 1H, ArH), 7.15 (m, 2H, ArH), 7.14 (d, *J* = 2.20 Hz, 1H, ArH), 6.99 (d, *J* = 6.10 Hz, 1H, ArH), 4.01–3.96 (m, 1H, CH), 3.77 (s, 3H, OCH_3_); 2.48 (d, *J* = 7.20 Hz, 2H, CH_2_), 1.92–1.81 (m, 1H, CH), 1.62 (d, *J* = 7.20 Hz, 3H, CH_3_), 0.92 (d, *J* = 6.60 Hz, 6H, CH_3_); ^13^C NMR (CDCl_3_) (δ[ppm]): 190.51 (C=O), 172.55 (C=O), 151.43, 144.25, 141.98, 140.76, 138.16, 137.17, 133.69, 132.77, 129.32, 128.61, 128.49, 127.39, 123.18, 122.24, 121.34, 111.89, 55.86 (OCH_3_), 44.97 (CH), 30.11 (CH), 22.38 (CH_3_), 18.62 (CH_3_).

2-Methoxy-4-(3-oxo-3-phenylprop-1-en-1-yl)phenyl-2-(4-benzoilphenyl)-propionate **(3aii)**

Yield: Colorless oil (37.32%), R_f_ = 0.49 (hexane/ethyl acetate, 2:1, *v*/*v*).

FT-IR (υ cm^−1^): 3322 (C-H), 3283 (C-H), 2927 (C-H), 2849 (C-H), 1758 (C=O), 1731 (C=O), 1658 (C=O), 1623 (C=C), 1575 (C=C), 1447 (C=C), 892 (=C-H), 776 (C-H); EI-MS, m/z (%): 490 (15) M^+^, 460 (100), 408 (2), 378 (9), 334 (7), 296 (2), 142 (1); ^1^H NMR (400 MHz, CDCl_3_) (δ[ppm]): 7.82–7.79 (m, 2H ArH), 7.78 (d, *J* = 15.40 Hz, 1H, =CH), 7.75 (m, 3H, CH), 7.70 (dt, *J* = 2.90, 1.70 Hz, 1H, CH), 7.68–7.55 (m, 2H, ArH), 7.54 (t, *J* = 6.50 Hz, 1H, ArH), 7.50 (m, 2H, CH), 7.47 (m, 3H, CH), 7.43 (d, *J* = 15.60 Hz, 1H, =CH), 7.38 (d, *J* = 11.50 Hz, 1H, ArH), 7.24 (dd, *J* = 8.29 Hz, *J* = 1.94 Hz, 1H, ArH), 7.14 (d, *J* = 1.90 Hz, 1H, ArH), 4.05 (m, 1H, CH), 3.78 (s, 1H, 3H, OCH_3_), 1.53 (d, *J* = 7.20 Hz, 3H, CH); ^13^C NMR (CDCl_3_) (δ[ppm]): 196.32 (C=O), 191.51 (C=O), 174.15 (C=O), 153.89, 146.38, 141.95, 140.95, 138.06, 137.87, 132.54, 132.45, 131.46, 131.26, 130.04, 129.20, 129.04, 128.91, 128.84, 128.75, 128.50, 128.31, 128.26, 121.30, 121.10, 112.75, 49.16 (OCH_3_), 45.42 (CH), 20.64 (CH_3_).

2-Methoxy-4-(3-oxo-3-phenylprop-1-en-1-yl)phenyl-2-(6-methoxynaphtalen-2-yl)-propioniate **(3aiii)**

Yield: White crystals (62.92%), m.p.: 102–103 °C, R_f_ = 0.41 (hexane/ethyl acetate, 9:2, *v*/*v*).

FT-IR (υ cm^−1^): 2980 (C-H), 2935 (C-H), 2850 (C-H), 1758 (C=O), 1663 (C=O), 1605 (C=C), 1509 (C=C), 1447 (C=C), 955 (=C-H), 775 (C-H), 741 (C-H); EI-MS, m/z (%): 466 (7) M^+^, 436 (100), 364 (16), 212 (2), 170 (5); ^1^H NMR (400 MHz, CDCl_3_) (δ[ppm]): 8.01–7.97 (m, 1H, ArH), 7.80 (d, *J* = 1.70 Hz, 1H, CH), 7.77 (s, 1H, CH), 7.74 (dd, *J* = 15.70, 9.30 Hz, 2H, CH_,_ =CH), 7.60–7.56 (m, 1H, ArH), 7.53 (d, *J* = 1.80 Hz, 1H, ArH_._), 7.50 (t, *J* = 7.40 Hz, 2H, ArH), 7.42 (d, *J* = 15.70 Hz, 1H, =CH), 7.19 (dd, *J* = 1.90 Hz, 1H, ArH), 7.17 (d, *J* = 2.60 Hz, 1H, CH), 7.15 (s, 1H, CH), 7.13 (d, *J* = 1.90 Hz, 1H, ArH), 6.98 (d, *J* = 8.20 Hz, 1H, ArH), 4.14 (m, 1H, CH), 3.93 (s, 3H, OCH_3_), 3.73 (s, 3H, OCH_3_), 1.71 (d, *J* = 7.10 Hz, 3H, CH_3_); ^13^C NMR (CDCl_3_) (δ[ppm]): 190.46 (C=O), 172.44 (C=O), 157.72, 151.55, 144.18, 141.93, 138.15, 135.11, 133.80, 132.79, 129.30, 128.97, 128.61, 128.48, 127.12, 126.37, 126.12, 121.16, 121.31, 118.99, 111.89, 105.64, 55.85 (OCH_3_), 55.30 (OCH_3_), 45.37 (CH), 18.68 (CH_3_).

2-Methoxy-5-(3-oxo-3-phenylprop-1-en-1-yl)phenyl-2-(4-isobutylphenyl)-propionate **(3bi)**

Yield: Light yellow needles (97.34%), m.p.: 99–100 °C, R_f_ = 0.60 (hexane/ethyl acetate, 2:1, *v/v*).

FT-IR (υ cm^−1^): 2961 (C-H), 2933 (C-H), 2874 (C-H), 1751 (C=O), 1657 (C=O), 1599 (C=C), 1574 (C=C), 1509 (C=C), 977 (=C-H), 772 (C-H), 691 (C-H); EI-MS, m/z (%): 443 (18) M^+^, 254 (13), 188 (100), 161 (40); ^1^H NMR (400 MHz, CDCl_3_) (δ[ppm]): 8.09–7.80 (m, 2H, ArH), 7.73 (d, *J* = 15.50 Hz, 1H, =CH), 7.60–7.42 (m, 3H, ArH), 7.36 (d, *J* = 15.50 Hz, 1H, =CH), 7.29 (d, *J* = 2.10 Hz 1H, ArH), 7.14 (dd, *J* = 8.20, 2.10 Hz, 1H, ArH), 6.69 (d, *J* = 8.60 Hz, 1H, ArH), 3.95 (s, 3H, OCH_3_), 3.64 (m, 1H, CH), 2.50 (t, *J* = 2.80 Hz, 2H, CH_2_), 1.93–1.86 (m, 1H, CH), 1.66 (d, *J* = 7.20 Hz, 3H, CH_3_), 0.95 (d, *J* = 1.90 Hz, 3H, CH_3_), 0.93 (d, *J* = 1.70 Hz, 3H, CH_3_); ^13^C NMR (CDCl_3_) (δ[ppm]): 186.30 (C=O), 173.53 (C=O), 153.23, 145.87, 143.87, 140.96, 140.79, 137.11, 132.67, 129.37, 128.69, 128.59, 127.44, 121.89, 120.71, 112.35, 110.5, 55.89 (OCH_3_), 45.08 (CH), 30.23 (CH_2_), 22.40 (CH_3_), 18.70 (CH_3_).

2-Methoxy-5-(3-oxo-3-phenylprop-1-en-1-yl)phenyl-2-(4-benzoilphenyl)-propionate **(3bii)**

Yield: White oil (91.67%), R_f_ = 0.54 (hexane/ethyl acetate, 2:1, *v*/*v*).

FT-IR (υ cm^−1^): 3320 (C-H), 3280 (C-H), 2925 (C-H), 2839 (C-H), 1750 (C=O), 1735 (C=O), 1653 (C=O), 1602 (C=C), 1510 (C=C), 955 (=C-H), 777 (C-H); EI-MS, m/z (%): 490 (16) M^+^, 460 (7), 335 (4), 209 (2), 104 (100), 97 (56); ^1^H NMR (400 MHz, CDCl_3_) (δ[ppm]): 7.80–7.78 (m, 2H, ArH), 7.72 (m, 3H, CH), 7.66 (dt, 1H, CH), 7.62 (d, *J* = 15.70 Hz, 1H, =CH), 7.59 (m, 3H, ArH), 7.45–7.51 (m, 2H, CH), 7.43 (m, 3H, CH), 7.33 (d, *J* = 15.60 Hz, 1H, =CH), 7.28 (d, *J* = 2.30 Hz, 1H, ArH), 7.18 (d, *J* = 2.60 Hz, 1H, ArH), 6.93 (d, *J* = 8.40 Hz, 1H, ArH), 4.04 (m, 1H, CH), 3.63 (s, 3H, OCH), 1.50 (d, *J* = 6.90 Hz, 3H, CH); ^13^C NMR (CDCl_3_) (δ[ppm]): 202.23 (C=O), 196.35 (C=O), 173.48 (C=O), 153.85, 153.23, 143.77, 141.89, 140.32, 138.00, 137.34, 132.61, 132.54, 131.26, 130.00, 129.02, 128.83, 128.74, 128.55, 128.43, 128.30, 121.82, 120.71, 112.33, 50.01 (OCH_3_), 45.28 (CH), 20.63 (CH_3_).

2-Methoxy-5-(3-oxo-3-phenylprop-1-en-1-yl)phenyl-2-(6-methoxynaphtalen-2-yl)-propioniate **(3biii)**

Yield: White solid (78.31%), m.p.:156 °C, R_f_ = 0.39 (hexane/ethyl acetate, 9:2, *v*/*v*).

FT-IR (υ cm^−1^): 3290 (C-H), 2930 (C-H), 2854 (C-H), 1697 (C=O), 1649 (C=O), 1604 (C=C), 1506 (C=C), 1036 (=C-H), 806 (C-H), 676 (C-H); EI-MS, m/z (%): 466 (6) M^+^, 253 (8), 185 (83), 169 (47), 153 (89), 141 (100), 115 (36), 104 (53), 77 (48); ^1^H NMR (400 MHz, CDCl_3_) (δ[ppm]): 7.99–7.96 (m, 2H, ArH), 7.81 (d, *J* = 1.80 Hz, 1H, CH), 7.76 (s, 1H, CH), 7.74 (d, *J* = 1.80 Hz, 1H, CH), 7.70 (d, *J* = 15.70 Hz, 1H, =CH), 7.57–7.46 (m, 3H, ArH), 7.45 (d, *J* = 1.80 Hz, 1H, CH), 7.42 (d, *J* = 1.60 Hz, 1H, CH), 7.33 (d, *J* = 15.60 Hz, 1H, =CH), 7.28 (d, *J* = 2.30 Hz, 1H, ArH), 7.18 (d, *J* = 2.60 Hz, 1H, ArH), 7.16 (s, 1H, CH), 6.93 (d, *J* = 8.40 Hz, 1H, ArH), 4.16 (m, 1H, CH), 3.93 (s, 3H, OCH_3_), 3.72 (s, 3H, OCH_3_), 1.72 (d, *J* = 7.10 Hz, 3H, CH_3_); ^13^C NMR (CDCl_3_) (δ[ppm]): 190.35 (C=O), 172.58 (C=O), 157.73, 153.23, 143.77, 140.32, 135.12, 133.82, 132.61, 129.30, 128.55, 128.43, 128.12, 127.14, 126.40, 126.29, 121.82, 120.71, 119.04, 112.33, 105.65, 55.89 (OCH_3_), 55.32 (OCH_3_), 45.35 (CH), 18.71 (CH_3_).

### 3.5. General Procedure Synthesis of Aromatic Aldehyde–NSAID Intermediate Derivatives (***2ai****–**2biii***)

Aromatic aldehyde **2a** or **2b** (5 mmol) was dissolved in dry CH_2_Cl_2_ (15 mL), followed by an anti-inflammatory drug (ibuprofen (**i**), naproxen (**ii**), or ketoprofen (**iii**) (6 mmol)), and then DCC (1.03 g, 8 mmol) and DMAP (0.05 g, 3 mmol) were added. The mixture was stirred at room temperature for 2 days. The resulting precipitate of dicyclohexylurea was filtered. The organic phase was washed with 5% aqueous HCl, 5% Na_2_CO_3_, and then with water. The organic layer was dried over the anhydrous MgSO_4_ and the solvent was removed under reduced pressure. The crude new esters (**2ai**–**2biii**) obtained were crystallised or purified by column chromatography on silica gel using a mixture of hexane/ethyl acetate (2:1, *v*/*v*) as an eluent.

4-Formyl-2-methoxyphenyl-2-(4-isobuthylphenyl)-propionate **(2ai)**

Yield: Colorless oil (44.85%), R_f_ = 0.40 (hexane/ethyl acetate, 2:1, *v*/*v*).

FT-IR (υ cm^−1^): 2953 (C-H), 2867 (C-H), 1758 (C=O), 1697 (C=O), 1601 (C=C), 1462 (C=C), 732 (C-H); EI-MS, m/z (%): 340 (32) M^+^, 297 (5), 188 (69), 161 (100), 117 (17).

5-Formyl-2-methoxyphenyl-2-(4-benzoilphenyl)-propionate **(2aii)**

Yield: Colorless oil (54.67%), R_f_ = 0.35 (hexane/ethyl acetate, 2:1, *v*/*v*).

FT-IR (υ cm^−1^): 2943 (C-H), 2835 (C-H), 1760 (C=O), 1690 (C=O), 1605 (C=C), 1505 (C=C), 725 (CH); EI-MS, m/z (%): 388 (54) M^+^, 236 (20), 209 (56), 151 (70), 105 (100), 77 (60).

4-Formyl-2-methoxyphenyl-2-(6-methoxynaphtalen-2-yl)-propionate **(2aiii)**

Yield: White solid (91.21%), m.p.:102–106 °C, R_f_ = 0.41 (hexane/ethyl acetate, 9:2, *v*/*v*).

FT-IR (υ cm^−1^): 2932 (C-H), 2850 (C-H), 1761 (C=O), 1699 (C=O), 1602 (C=C), 1502 (C=C), 734 (C-H); EI-MS, m/z (%): 364 (100) M^+^, 258 (2), 185 (94), 141 (8), 115 (2).

5-Formyl-2-methoxyphenyl-2-(4-isobutylphenyl)-propionate **(2bi)**

Yield: Colorless oil (54.62%), R_f_ = 0.54 (hexane/ethyl acetate, 2:1, *v*/*v*).

FT-IR (υ cm^−1^): 2953 (C-H), 2868 (C-H), 2843 (C-H), 1759 (C=O), 1688 (C=O), 1606 (C=C), 11,457 (C=C), 744 (C-H); EI-MS, m/z (%): 340 (3) M^+^, 299 (4), 257 (2), 239 (6), 187 (55), 162 (73), 161 (100), 117 (22).

5-Formyl-2-methoxyphenyl-2-(4-benzoilphenyl)-propionate **(2bii)**

Yield: Colorless oil (78.43%), R_f_ = 0.31 (hexane/ethyl acetate, 2:1, *v*/*v*).

FT-IR (υ cm^−1^): 2935 (C-H), 2825 (C-H), 1759 (C=O), 1688 (C=O), 1606 (C=C), 1510 (C=C), 718 (C-H); EI-MS, m/z (%): 388 (21) M^+^, 236 (36), 209 (59), 151 (77), 105 (100), 77 (76), 51 (27).

5-Formyl-2-methoxyphenyl-2-(6-methoxynaphtalen-2-yl)-propionate **(2biii)**

Yield: White solid (84.62%), m.p.:177 °C, R_f_ = 0.62 (hexane/ethyl acetate, 9:2, *v*/*v*).

FT-IR (υ cm^−1^): 3290 (C-H), 2930 (C-H), 2854 (C-H), 1697 (C=O), 1649 (C=O), 1605 (C=C), 1505 (C=C), 744 (C-H); EI-MS, m/z (%): 364 (3) M^+^, 311 (4), 211 (34), 185 (100), 184 (60), 170 (39), 141 (28), 115 (15).

### 3.6. General Synthesis Procedure of Curcuminoid Chalcone–NSAID Hybrid Derivatives (***3ai****–****3biii***) in the Reaction of Aromatic Aldehyde–NSAID Intermediate Derivatives (***2ai****–****2biii***) with Acetophenone

The aqueous solution of 50% NaOH (5 mL) was added dropwise into a solution of 3 mmol (0.36 g) of acetophenone in ethanol (20 mL). A total of 3 mmol of vanillin–NSAID ester (**2ai**–**iii**) or isovanillin–NSAID ester (**2bi**–**iii**) in an ethanolic solution (5 mL) was introduced into the mixture. The mixture was stirred at room temperature for 48 h. Then, the mixture was poured into ice water and neutralised with 10% HCl to produce precipitates. The formed solid product was filtered, washed with water, and crystallised with methanol to yield the final compounds. The aquatic solution was extracted with dichloromethane when the solid was not formed. The combined organic layer was washed with 10% HCl and then with water. The organic layer was dried over the anhydrous MgSO_4_ and the solvent was removed under reduced pressure. The crude solid was purified by column chromatography using hexane/ethyl acetate 2:1 (*v*/*v*) as an eluent.

As a result of the synthesis, compounds **3ai**–**3biii** were obtained. The compounds’ identities were consistent with the products obtained according to the general synthesis procedure for curcuminoid chalcone–NSAID hybrid derivatives in the reaction of curcuminoid chalcones with NSAIDs.

### 3.7. Stability Tests of Hybrid Curcuminoid Chalcones

#### 3.7.1. Apparatus and Chromatographic Conditions

The HPLC-UV chromatographic analysis was performed using an Agilent 1220 Infinity LC System chromatograph (Böblingen, Germany). A Luna^®^ 3 µm C18(2) 100 Å, 150 × 4.6 mm column (Phenomenex, Shim-Pol A.M. Borzymowski, Izabelin, Poland) was used as a stationary phase. The mobile phase was a mixture of acetic acid (20 mM) and NaCl (1 mM)-acetonitrile (20:80, *v*/*v*). The column temperature was 30 °C and the autosampler temperature was 25 °C. The flow rate of the mobile phase was 1.0 mL/min and the injection volume was 10 µL. The DAD detection was performed at 254, 308, and 340 nm. The Atlas Suntest CPS+ (Atlas, Linsengericht—Altenhasslau, Germany) and also the thermostat ST-1+ (Pol-Eko, Wodzisław Śląski, Poland) were used to expose the substances in highly accelerated and stress testing. The stress tests were performed in a KBC-100 thermal testing chamber (Wamed, Warsaw, Poland) with a heat set temperature accuracy of ±1 °C. The other equipment used was a FBH 612 ultrasonic bath (Fischerbrand, Schwerte, Germany). The IR spectra were recorded using a IRAffinity-1 spectrophotometer (Shimadzu, Kioto, Japan).

#### 3.7.2. Stress Test

##### Strong Acid, Strong Base, and Neutral Degradation

Compound **2** or compound **6** (1.0 mg) was dissolved in 1.5 mL of a mixture of dimethyl sulfoxide (DMSO) and methanol (MeOH) (1:6, *v*/*v*). Then, 2.0 mL samples of the tested solutions (c ~60 g/mL) were prepared in hydrochloric acid (0.1; 0.01 M) or sodium hydroxide solution (0.1, 0.01 M) and demineralised water. The solutions were transferred to glass vials, encapsulated, and stored in a thermal chamber at 90 °C, 40 °C, or 25 °C. After the appropriate time (Table 1), the samples were cooled, diluted with methanol (1:1), and transferred to the chromatography column.

##### Oxidative Degradation

Stock solutions of the test substances (**2** or **6**) were prepared by dissolving 1 mg of each test substance in 1.5 mL of a mixture of DMSO and MeOH (1:6, *v*/*v*). Then, 2 mL solutions of compounds **2** and **6** were prepared at ~30 μg/mL in a hydrogen peroxide solution (1, 3, and 10%). The solutions were placed in dark glass vials and encapsulated, and after an appropriate time (Table 1) the samples were diluted with methanol (1:1, *v*/*v*) and transferred to the chromatographic column.

##### Photostability of Compound **2** and Compound **6** in the Solutions

Stock solutions of the test substances (**2** or **6**) were prepared by dissolving 1 mg of each test substance in 1.5 mL of a mixture of DMSO and MeOH (1:6, *v*/*v*). Then, 5 mL solutions of the tested compounds (c ~ 15 g/mL) in MeOH were prepared. Next, 2.5 mL samples of the solutions were transferred to quartz cuvettes and tightly closed; one of the cuvettes was packed in aluminium foil, and both were irradiated for 22 h in the Atlas Suntest CPS+ instrument (250 W/m^2^; 1.2 × 10^6^ lux·h). Each of the samples (protected and irradiated) was transferred to the chromatographic column. Samples before irradiation were also subjected to chromatographic analysis (t = 0) (Table 2).

##### Photostability of Compound **2** and Compound **6** in the Solid State

The two test compounds, **2** and **6** (1 mg), were prepared with KBr (300 mg) in tablet form. Each tablet was placed for 110 h in the Atlas Suntest CPS+ instrument (250 W/m^2^; 6.0 × 10^6^ lux·h). A tablet exposed to light (unprotected) and protected from light (with aluminum foil) was prepared for each test substance. IR spectra of the substances tested before and after irradiation were taken (Table 2).

##### Stability Test in Release Solutions

Stock solutions of the test substances (**2** or **6**) were prepared by dissolving 2.5 mg of each test substance in 0.7 mL of a mixture of DMSO and MeOH (1:6, *v*/*v*). At the same time, 7.8 mL of the HCl solution (0.1 M, pH 1.2) in 1% Tween 80 and release buffer at pH 6.8 in 1% Tween 80 were prepared. The solutions were thermostatted for approximately 30 min in a water bath at 37 °C and then 0.2 mL of the test compound solution (c ~70 µg/mL) was added to each. At appropriate intervals, 0.5 mL of the thermostatted solution was taken, diluted with 0.5 mL methanol, and transferred to the chromatographic column.

### 3.8. Sonodynamic Method

#### Cell Culture and Viability Cells

SCC-25 and FaDu human squamous cell carcinoma cell lines were purchased from the American Type Culture Collection (ATCC). The cells were cultivated in an incubator under optimal culture conditions (temperature: 37 °C; humidity: 95%; carbon dioxide content: 5%). The SCC-25 cell line was cultured in phenol red-free DMEM:F-12 medium complemented with 10% FBS, 2.5 mM L-glutamine, penicillin (100 U/mL), and streptomycin (0.1 mg/mL). The FaDu cells were maintained in phenol red-free Eagle’s Minimum Essential Medium (EMEM) supplemented with 10% fetal bovine serum (FBS), 2.5 mM L-glutamine, penicillin (100 U/mL), and streptomycin (0.1 mg/mL). For the experiments, confluent stock cultures were harvested using the trypsin–EDTA solution and seeded in 35-mm-diameter Petri plates at a density of 1 × 10^6^. They were allowed to attach overnight, and the tested compounds were then added at a concentration of 40 µM. Control cells were maintained under the same conditions with 0.1% DMSO. After 8 h of incubation, both control and tested cells were exposed to the US for 3 min using a 35 mm transducer with a resonance frequency of 3 MHz and ultrasonic intensity of 0.8 W/cm^2^. The MTT test was performed after 6 h of sonotherapy to assess the viability of the cells. The suspensions were removed and replaced with an MTT/EMEM or MTT/DMEM:F-12 mixture (1:8). The resulting formazan crystals were dissolved by adding 1 mL of DSMO. The absorbance was measured using an Elx-800 plate reader (BioTek, Bad Friedrichshall, Germany) at 570 nm (reference wavelength 650 nm).

### 3.9. Estimation of Antineurodegenerative, Anti-Inflammatory, and Antioxidant Activity

#### 3.9.1. Sample Preparation

The samples were dissolved in DMSO (Sigma D4540) to obtain a concentration of 3 mM/dm^3^.

#### 3.9.2. Effect on AChE and BChE Activity

The analysis was performed strictly as described by Studzińska-Sroka et al. [42], except that the volume of the tested samples added to the reaction mixture was 0.035 mL.

#### 3.9.3. Effect on COX-2 Activity

The analysis was performed strictly as described by Studzińska-Sroka et al. [42], except that the volume of the tested samples added to the reaction mixture was 0.02 mL.

#### 3.9.4. Effect on GPx and GR Activity

The analyses were performed strictly as described by Studzińska-Sroka et al. [42].

#### 3.9.5. Effect on CAT Activity

The method used by Watanabe et al. [44] was used with the following modifications. The reaction mixture was composed of 0.02 mL of EDTA solution (56.5 mM), 0.01 mL of ^the^ sample, 0.02 mL of 3% H_2_O_2_ solution (Sigma H1009), and 0.02 mL of catalase solution (4000-fold diluted, Sigma C3515) (all reagents except the tested sample were diluted in TRIS buffer, pH 7.0, 1 mol/dm^3^). The volume was completed to 0.31 mL by the same buffer solution. In a blank sample, DMSO (Sigma D4540) replaced the tested sample. The background of the sample was measured in a mixture composed of 0.01 mL of the sample completed to 0.31 mL by the buffer. The absorbance was read at 240 nm directly after the mixing and after 5 min of incubation (room temperature). The decreased absorbance (depletion of H_2_O_2_) in the tested and blank samples was compared. The calibration curve was produced using eleven H_2_O_2_ solutions (0.5693–5.693 mM/dm^3^). The results are expressed in % inhibition and as H_2_O_2_ depletion (mmol of depleted H_2_O_2_/dm^3^ min).

#### 3.9.6. HORAC

An analysis was performed as described by Szwajgier et al. [45]. The concentration of the samples was 1 mg/mL DMSO. The only modification was that the volumes of all reagents used for the measurements were reduced 4-fold (proportionally) to measure the absorbance using a microplate reader (Varioskan Lux, Thermo Scientific).

## 4. Conclusions

In summary, convenient methods have been developed to synthesise curcuminoid chalcone–NSAID hybrids via two methods. In the reactions of appropriate aromatic aldehydes with acetophenone in an alkaline condition, curcuminoid chalcones were obtained, which as substrates were used in further transformations to obtain new ester-type compounds. The structures of the new compounds obtained have been confirmed using FT-IR, MS, and NMR methods.

The stress tests, photostability analysis in solutions, and solid phase of the two ester-type hybrid compounds with ibuprofen in the tests marked as **2** and **6** were determined according to the International Conference on Harmonization guidelines. Both compounds are characterised by a similar susceptibility to degradation factors, i.e., they are extremely unstable in alkaline conditions, very unstable in acidic conditions, unstable in neutral conditions, practically stable in oxidising conditions, and photolabile in solutions and in the solid state. Structural differences, however, affect the differences in their stability. Therefore, the isomer, in which the ester group was in the meta position of the curcuminoid chalcone, was slightly more stable in the tested conditions (i.e., stress test and stability test in release solutions). These tested compounds maintain adequate stability in pH 1.2 and 6.8 conditions, which may make them good candidates for developing formulations for oral administration.

The transformation of curcuminoid chalcones to afford hybrid ester-type derivatives in the reaction of the hydroxy group in the chalcone skeleton with the free carboxylic moiety of anti-inflammatory molecules allowed for obtaining promising anticancer agents within the studied cell lines.

Combining the curcuminoid chalcone based on vanillin with ibuprofen has shown the strongest potential anticancer effect on the FaDu cell line. In addition, the biological study confirmed that anticancer activity is an important aspect of hydroxy and methoxy groups in at least one aromatic ring of curcuminoid chalcones.

Our study showed that the sensitivity of SCC-25 and FaDu cells to the compounds marked as **2** and **3** is significantly greater when treated with a combination of the tested compounds and the ultrasounds. Compounds **2** and **3** showed promising anticancer potential in the studied cell line, and these compounds could be effective sonosensitisers for sonodynamic therapy.

The in vitro tests confirmed the high antineurodegenerative, anti-inflammatory, and antioxidant activity levels of the curcuminoid chalcones and curcuminoid chalcone–NSAID hybrids. The results exhibited the excellent anticholinergic activity of all of the tested compounds, with potential applications for AD treatment. Selected compounds based on the chalcone structure, especially compound **7**, exerted anticholinesterase activities and anti-COX-2 activities in the in vitro tests. Note that the curcuminoid chalcone–NSAID hybrids **2** and **6** inhibited AChE and BChE more than their substrates **1** and **7**. The compounds tested, especially compound **7**, very efficiently reduced the activity of the inflammatory enzyme COX-2. Additionally, the curcuminoid chalcones **1** and **7** showed a strong effect on the so-called “antioxidant” enzymes (GPx, GR, and CAT). The curcuminoid chalcone–NSAID hybrids **2** and **6** showed antioxidant activity when evaluated in the HORAC test.

In conclusion, curcuminoid chalcone derivatives have been shown to be very active compounds as new neuroprotective and sonosensitising agents.

## Data Availability

Data is contained within the article.

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
