# Peer review of "Curcuminoid Chalcones: Synthesis, Stability, and New Neuroprotective and Sonosensitising Activities"

_pharmaceuticals, 2023, doi:10.3390/ph16091331_

Round 1

Reviewer 1 Report

Please find my comments below:

·       I think that the volume of these parts in Discussion/Results chapter (2.2.3. Anticholinergic activity) could be a little reduced as they look more like Introduction and NOT like the discussion with other authors. This refers more or less to 2.2.3. as well as some other subsections

·       "Alzheimer’s disease" - I propose to use an abbreviation, because this term appears many times in the text.

·       I suggest improving Scheme 1. Substrates at the reaction arrow should be used in the singular (NSAIDs not NSAIDs). Furthermore, in the general formulas of the final compounds 2ai-2aiii, 2bi-2biii and 3ai-3aiii, 3bi-3biii the term NSAID (used in the blue circle) is not entirely correct. I suggest changing to NSAID-moiety or NSAID-group.

·       In the chapter 2.1.1. the term "NSAD" in the title should be corrected to "NSAID"

·       The term "in vitro" should consistently be written in italics throughout the paper

·       I think the authors should include NMR spectra in Supplementary data.

·       In the experiential part: sentences should not start with a number.

·       Lines 777, 785, 788 and others: The ratio in which the solvents were mixed is given. Was it by weight or volume ratio?

·       What does the sentence mean? Specifically, the information in parentheses: Then, 2 mL of compound 2 or compound 6 solutions (c ~30 g/mL) were 785 prepared in hydrogen peroxide (H2O2: 1, 3, 10%).

·       What does "applied to the chromatography column" mean? Was the reaction carried out on the column? Has the product been purified? If purified, information on the eluent is missing.

Author Response

Pharmaceuticals-2609245

Curcuminoid Chalcones; Synthesis, Stability, New Neuroprotective and Sonosensitising Activity

We would like to thank you for your kind comments and constructive criticism. Please find all comments addressed below in the text. All changes are highlighted in red colour.

According to the suggestion of the Reviewer, all necessary corrections have been made according to the instructions. We are very grateful to the Reviewer for taking the time to review our manuscript.

Reviewer 2 Report

The manuscript reports the synthesis of Curcuminoid Chalcones and their pharmacological activities. The manuscript is interesting because it reports the synthesis of new hybrid chalcones containing anti-inflammatories that demonstrated excellent biological activities. However, before being accepted it is necessary to make some corrections:

1) Why do authors use the term curcuminoid to refer to chalcones?

2) Why did the authors choose ibuprofen derivatives to carry out the biological activities?

3) Add to the manuscript the proposed fragmentation of the obtained derivatives.

4) Degradation tests indicated that ibuprofen derivatives are degraded in acidic or basic media, the degradation products were not identified. So what is the purpose of the synthesis since when ingested as drugs, the compounds would be degraded by the gastric juice and would lose their activity. How would the authors solve this problem and make the drug viable?

5) Toxicity tests must be performed by the authors.

6) What is the degree of purity of the synthesized compounds? Add the chromatograms of the synthesized compounds

7) Which eluent is used in the flash column to purify the compounds

8) English must be improved

8) English must be improved

Author Response

(The authors gave the same response as above.)

Round 2

Reviewer 1 Report

I don't have more questions.

Reviewer 2 Report

The manuscript presents a great contribution to the field and in my opinion can be accepted for publication